# Evolution of Youth’s Mental Health and Quality of Life during the COVID-19 Pandemic in South Tyrol, Italy: Comparison of Two Representative Surveys

**DOI:** 10.3390/children10050895

**Published:** 2023-05-17

**Authors:** Verena Barbieri, Christian J. Wiedermann, Giuliano Piccoliori, Angelika Mahlknecht, Barbara Plagg, Dietmar Ausserhofer, Ulrike Ravens-Sieberer, Adolf Engl

**Affiliations:** 1Institute of General Practice and Public Health, Claudiana College of Health Professions, 39100 Bolzano, Italy; 2Department of Public Health, Medical Decision Making and Health Technology Assessment, University of Health Sciences, Medical Informatics and Technology, 6060 Hall, Austria; 3Faculty of Education, Free University of Bolzano, 39100 Bolzano, Italy; 4Department of Child and Adolescent Psychiatry, Psychotherapy, and Psychosomatics, University Medical Center Hamburg-Eppendorf, 20251 Hamburg, Germany

**Keywords:** anxiety, COVID-19, depression, health-related quality of life, home schooling, Italian children and adolescents, lockdown, longitudinal study, mental health, pandemic, psychosomatics

## Abstract

Background: The coronavirus disease 2019 pandemic has led to an increase in youth mental health problems worldwide. Studies have revealed substantial variation in the incidence of these problems across different regions. Longitudinal studies of children and adolescents in Italy are lacking. This study aimed to investigate the development of health-related quality of life (HRQoL) and mental health in Northern Italy by comparing surveys conducted in June 2021 and in March 2022. Methods: A representative, large cross-sectional, online survey investigated HRQoL, psychosomatic complaints, and symptoms of anxiety and depression among 5159 and 6675 children and adolescents in 2021 and 2022, respectively, using the KIDSCREEN-10 index, HBSC symptom checklist, SCARED, CES-DC, and PHQ-2 instruments. Statistical analyses included a multivariate linear regression analysis. Results: Baseline characteristics showed significant differences in demographic variables between the two surveys. Girls and their parents reported a significantly lower HRQoL in 2021 than in 2022. Psychosomatic complaints differed significantly between sexes, and the results showed no decrease in psychosomatic complaints, anxiety, or depression between 2021 and 2022. Predictors of HRQoL, anxiety, depressive symptoms, and psychosomatic complaints in 2022 differed from those in 2021. Conclusions: The characteristics of the 2021 pandemic, including lockdowns and home schooling, may have contributed to the differences between the two surveys. As most pandemic restrictions ended in 2022, the results confirm the need for measures to improve the mental and physical health of children and adolescents after the pandemic.

## 1. Introduction

The global prevalence of anxiety and depression symptoms in children and adolescents has experienced a notable increase during the course of the COVID-19 pandemic, with a particularly significant rise observed from the beginning of the pandemic in 2020 to 2021 [1]. Persistent changes in daily routines during the pandemic period, affecting young people’s social interactions and schools, are likely to have led to the observed increase in mental health problems. Despite the initial decrease in psychiatric specialist consultations during the pandemic among children and adolescents due to lockdown measures, more psychiatric treatments during the pandemic were associated with an increase in mental health problems [2], including suicidal or self-harm ideation [3,4,5] and psychotic-like experiences [6].

In a recent systematic review, methodological heterogeneity among the studies limited the ability to conduct meta-analyses. Although the results confirmed worsening mental health and health-related quality of life (HRQoL) in children and adolescents, they were heterogeneous without a clear discernible pattern [7]. However, because even minor psychosocial and mental health deteriorations in children and adolescents can have significant implications for care at the population level, scientific improvements in the database are important.

Studies have compared the mental health problems of youth not only during the pandemic with those before the pandemic but also longitudinally within the different phases as the pandemic progressed. The mental and behavioral effects of the pandemic were independent of whether the surveys were conducted in or out of lockdown [8,9,10]. For anxiety symptoms of youth during the pandemic, which slowed in increase after the first 5 months, a slight downward trend was observed from the winter semester of 2021 to the fall of 2022 [11]. Home schooling established over time and the withdrawal of lockdown measures may have led to the recovery or stabilization of youth mental health. Further research is needed to assess how this downward trend in anxiety disorders develops over time, as well as how other areas of mental health and HRQoL develop, since pandemic-associated anxiety is an identified risk factor for increased symptoms of depression [12].

Globally, there has been a marked increase in mental health problems as a result of the pandemic, and geographic variation in incidence appears to be substantial [11]. There are a number of informative longitudinal and repeat surveys in addition to many snapshots of individual cross-sectional surveys; however, observations over longer time periods are limited and reflect heterogeneities, particularly in Europe [13]. After an initial observation of a reduction in HRQoL in Southern Germany [14], a nationwide longitudinal study collected data for 3 years since the start of the pandemic and found that the prevalence of mental health problems and anxiety increased and HRQoL decreased from 2020 to 2022 compared with pre-pandemic levels [15]. Similar studies on the increased prevalence of mental and physical impairments due to the pandemic have been reported recently in Germany [16] and other European countries, including Norway [12,17,18], Denmark [8], the United Kingdom [19], France [20], and Switzerland [21,22]. In a longitudinal study in the United Kingdom, only depressive symptomatology was significantly more common than pre-pandemic, but anxiety and emotional problems were not [23]. A Dutch study of adolescents in an at-risk group with mental health problems before the pandemic failed to find an increase in symptoms during the pandemic in this subgroup [24]. Corresponding observations have been reported in Switzerland regarding the wellbeing of children with complex medical histories [25].

In Italy, longitudinal mental health data before and during the pandemic have been reported for students aged 18–30 years [26] and youths with attention deficit hyperactivity disorder [27]; however, meaningful longitudinal studies of the general population below 18 years of age are largely lacking. Surveys on the mental health, HRQoL, and wellbeing of children and adolescents during the pandemic are only available as cross-sectional studies characterized by a variety of limitations, making it difficult to draw valid conclusions about the actual psychosocial impact of the pandemic on youth mental health in Italy.

In South Tyrol, Italy, a large cross-sectional population-based study assessed HRQoL and the mental health of children and adolescents in 2021, the second year of the COVID-19 pandemic [28]. This survey was replicated in 2022 with identical designs and target populations, and the results are reported herein.

This study aimed to identify regional factors and cultural contexts that may influence mental health outcomes in Northern Italy, enhancing the understanding of the pandemic’s impact on youth mental health in diverse settings. The survey replication comparatively investigated how the HRQoL and mental health of children and adolescents continued to develop in the third year of the pandemic in South Tyrol after all schools were opened again, restrictions were relaxed, and anxiety-reducing activities were experienced with coronavirus vaccinations and infections. The objective was to show whether and which psychosocial problems have improved, as well as whether groups of children are still more at risk than others are. The following research questions were addressed:How did mental health symptoms, especially anxiety, depression, and psychosomatic complaints, develop during the pandemic?Do longitudinal data only show the development of mental health symptoms, or can we derive information about changes in predictors over time?What are the differences between proxy-reported age groups (7–10 years and 11–19 years) in the longitudinal development of mental health problems?Are there groups more vulnerable to mental health problems at the local level?

We hypothesized that (i) HRQoL would exhibit significant changes with the evolving conditions of daily life during the pandemic, such as lockdowns and homeschooling, (ii) mental health problems, particularly symptoms of anxiety and depression, persist as long-lasting issues, even after the relaxation of restrictive measures, (iii) longitudinal data provide insight into changes in predictors over time, (iv) there would be notable differences between proxy-reported age groups (7–10 years and 11–19 years) in the longitudinal development of mental health problems, and (v) specific subgroups within the population exhibit increased vulnerability to mental health problems at the local level.

By comparing data from 2021 and 2022, we aimed to understand the extent to which mental health problems, which have intensified during the pandemic, will require ongoing monitoring and targeted interventions in the coming years.

## 2. Materials and Methods

### 2.1. Study Design and Sample

The study design, sample, and type of public school included in the survey of HRQoL and mental health of children and adolescents in South Tyrol were previously described [28]. Repeated cross-sectional “Corona and Psyche in South Tyrol” (COP-S) 2021 and 2022 studies were conducted during the second and third years of the pandemic. Surveys were carried out among families with children and adolescents aged 7–19 years in South Tyrol. The surveys included parental ratings and self-rated questionnaires for children and adolescents aged 11–19 years. The figure below shows the data collection timepoints, *n* for each data collection, and the related pandemic phases in Central Europe (see Figure 1).

The methodology was similar to that of COPSY German studies [15,29,30]. The study was approved by the local ethics committee, which also approved the parents’ consent and privacy declaration. The two study waves were anonymous online surveys targeting families with children aged 7–19 attending public schools in South Tyrol. In collaboration with the public schools’ administrations, around 38,400 families received invitations via email to participate in the survey. The email contained a link leading to the study description, where parents were asked for their informed consent and children for their assent. A reminder invitation was sent after 1 week to encourage participation. The survey assessed sociodemographic variables, COVID-19 burden, HRQoL, and mental health. The required sample size was calculated using G-Power software, considering a cutoff for a sample size of alpha = 0.05, a power of 0.9, a small effect of 0.1, and 14 predictors, resulting in a minimum required sample size of 243 as previously described [28].

Of the 6957 and 9171 returned questionnaires, 1234 (17.7%) and 1612 (17.6%) were excluded because parents did not agree with the privacy declarations, 468 (6.8%) and 745 (8.1%) only filled in the information about themselves, but not about their child, and 86 (1.2%) and 149 (1.6%) were excluded because they were identified as double entries, respectively. As described in [28], for comparisons between parents and adolescents, only the questionnaires answered by both sides were included.

### 2.2. Measures

#### 2.2.1. Sociodemographic Variables and COVID-19 Burden

Sociodemographic variables of parents and children were collected, including questions on marital status, migration background, occupational status, and parental education, according to the “Comparative Analysis of Social Mobility in Industrial Nations” (CASMIN) index [31,32].

Parents were asked about the overall burden as well as the work-related burden caused by the pandemic, using items developed for the COPSY Germany 2020 questionnaire [29,30]. Questions about pandemic-related burdens regarding family climate, social distancing, use of digital media, school closure, nutritional behaviors, and changes in sports behaviors were directed toward both parents and youth.

#### 2.2.2. HRQoL and Mental Health

The details of the psychometric instruments were described elsewhere [28] and are summarized here. HRQoL was assessed using the KIDSCREEN-10 index, developed as a valid instrument for children and adolescents consisting of ten questions on physical, psychological, social, and school-dependent items presented on a five-point response scale [33]. The Screen of Child Anxiety Disorders (SCARED) [34], with nine items on symptoms of generalized anxiety represented on a three-point response scale, was used as a psychometrically validated instrument for German [35] and Italian [36] children and adolescents. 

The presence and frequency of different psychosomatic problems in children and adolescents were assessed by parents and adolescents using the adopted Health Behavior in School-Aged Children (HBSC) symptom checklist [37] within the last week, with items presented on a five-point response scale for quantification and comparison with the COP-S study results from 2021.

The two item Patient Health Questionnaire-2 (PHQ-2) [38] on a four-point Likert scale (0–3) was used as a depression screening tool, with a sum score of 3 as the cutoff. The PHQ-2 fulfills important psychometric criteria [39,40].

#### 2.2.3. Data Analysis

As previously described [28], the data were analyzed using descriptive statistics, using the mean (M) and standard deviation (SD) for metric variables, and the absolute and relative frequencies for nominal- and ordinal-scaled variables. Chi-square tests were performed to compare categorical variables, along with Kendall’s tau-b as correlation coefficients. A comparison of parents and children’s responses was performed only for the subgroup of 11–19 year old scholars. Low HRQoL and HBSC subscales were compared using McNemar’s test for paired variables. Significance levels of α < 0.05 *, α < 0.01 **, and α < 0.001 *** are reported. 

Multivariate linear regression analyses were performed for all summed scores to account for sociodemographic and pandemic-specific risk factors. In proxy-reported surveys conducted in 2021 and 2022, younger age, no single parenthood, absence of elevated pandemic-related parental burden, good family climate, absence of elevated school burden, moderate use of digital media, and good contact with friends were significant predictors of better HRQoL. Different predictors were identified in the self-reported HRQoL surveys. In the 2021 self-reported survey, better HRQoL was predicted by female × younger age, absence of elevated parental burden due to the pandemic, good family climate, absence of elevated school burden, moderate use of digital media, and good contact with friends. For better self-reported HRQoL in the 2022 survey, younger age × female sex, good family climate, no elevated burden of school, and moderate use of digital media were significant predictors. For an increase in self-reported symptoms of anxiety in 2021, older age × female gender was a significant predictor, as well as elevated parental burden due to the pandemic, lower family climate, elevated school burden, and less contact with friends. In 2022, increased anxiety, according to adolescents’ self-reports, was identified as the same predictor. The significant predictors of increased depressive symptoms self-reported by adolescents in 2021 included older age, older age × female sex, elevated parental burden due to the pandemic, a lower family climate, elevated school burden, and less contact with friends. In comparison, in 2022, elevated parental burden due to the pandemic was no longer predictive. Stronger differences were found in predictors of increased self-reported psychosomatic symptoms. In 2021, older age was a significant predictor, as was older age × female sex, elevated parental burden due to the pandemic, lower family climate, and elevated school burden. In 2022, older age, older age × female sex, medium/high parental education, lower family climate, less contact with friends, and an elevated school burden were significant predictors. Increased proxy-reported psychosomatic complaints in 2021 were significantly dependent on older age, older age × female sex, single parenthood, elevated parental burden due to the pandemic, lower family climate, and elevated school burden. In 2022, in addition to the 2021 factors, less contact with friends and migration background were significant predictors.

Regression diagnostics were checked for the independence and normality of residuals, homoscedasticity, multicollinearity (variance inflation factor reported), and autocorrelation of error terms (Durbin–Watson statistics reported). The outliers were identified using Cook’s distance. All statistical analyses were performed using SPSS version 25.0 (Armonk, NY, USA).

## 3. Results

### 3.1. Baseline Characteristics

The study followed a two-timepoint design, with the baseline assessment conducted in June 2021 and the follow-up assessment in March 2022. Results are shown in Table 1. Analysis of the demographic variables of the participants showed significant differences in baseline characteristics between June 2021 and March 2022 for mean age of children (12.0 vs. 12.21; *p* < 0.001) and parents (44.4 vs. 44.7; *p* < 0.01), as well as low educational standards of parents (23.0% vs. 20.6%; *p* < 0.01).

The age and gender of children and adolescents, migration background, single parenthood, and age of parents are representative of the South Tyrolian population, according to ASTAT [https://astat.provinz.bz.it/de/datenbanken-gemeindedatenblatt.asp (accessed on 15 May 2023)].

### 3.2. Health-Related Quality of Life (HRQoL)

The parents and adolescents reported their HRQoL. Table 2 shows the percentage of participants up to 11 years of age with a low HRQoL. Girls (low HRQoL: 38.0% vs. 33.6%, *p* = 0.033) and their parents (33.5% vs. 26%, *p* < 0.001) reported significantly lower HRQoL in 2021 than in 2022, whereas boys reported no significant change from 2021 to 2022. Boys’ parents reported significantly lower HRQoL in 2021 (30.5% vs. 25%, *p* < 0.001).

Girls reported significantly lower HRQoL than boys in both 2021 and 2022, whereas parents reported no significant differences between boys and girls. Parents reported significantly (*p* < 0.001) lower HRQoL in both surveys than their children did.

### 3.3. Psychosomatic Complaints (HBSC)

Psychosomatic complaints were presented separately for children (7–10 years) and adolescents (11–19 years). Data were proxy-reported by parents and self-reported by the adolescents.

In self-reports by adolescents (11–19 years), boys reported significantly less often than girls about weekly headache (32% vs. 47% in 2021, and 36.3% vs. 51.1% in 2022, respectively), abdominal pain (23.5% vs. 40.3% and 26.7% vs. 41.7%, respectively), backache (25.9% vs. 32.4% and 25.5% and 33.1%, respectively), feeling low (34.6% vs. 48.5% and 35.0% vs. 50.8%, respectively), nervousness (43.4% vs. 47.9% and 40.8% vs. 42.6%, respectively), irritability (55.7% vs. 66.8% and 52.4% vs. 63.9%, respectively), sleeping difficulties (38.8% vs. 45.0% and 36.8% vs. 43.7%, respectively), and dizziness (15.9% vs. 25.9% and 16.8% vs. 25%, respectively).

For younger children (7–10 years old), only proxy reports were available, and few differences were found between boys and girls. Parents reported significantly more nervousness for boys than for girls in 2021 (37.1% vs. 30.8%) and 2022 (35% vs. 28.5%), and they reported significantly more sleep problems for girls in both survey years: 2021 (34.1% vs. 38.8%) and 2022 (26.9% vs. 32.7%). They also reported a significantly higher prevalence of weekly headaches (22.6% vs. 28.6%) and abdominal pain (28.0% vs. 34.2%) in girls in 2022.

For adolescents (11–19 years), parents did not report differences between boys and girls in feelings of nervousness in either year. For all other complaints, parents reported a significantly higher percentage of girls than boys. Details are presented in Table 3.

Comparisons between the 2021 and 2022 self-reported HBSC data showed a significant increase in headaches in boys (32% vs. 36.3%, *p* < 0.045). 

Parents’ proxy reports for adolescents indicated a significant increase in headaches between 2021 and 2022 for both boys (28.2% vs. 31.9%, *p* = 0.031) and girls (39.4% vs. 45.8%, *p* < 0.001), and a significant decrease in irritability again for both boys (64.5% vs. 59.1%, *p* < 0.003) and girls (69.5% vs.65.9%, *p* < 0.04).

Parents’ proxy reports for children aged 7–10 years showed a significant increase in weekly headaches (22.9% vs. 28.6%, *p* = 0.013) and abdominal pain (29.6% vs. 34.2%, *p* = 0.036), and a significant decrease in weekly sleeping problem frequencies (38.8% vs. 32.7%, *p* < 0.007) in girls. Boys’ parents reported a significant increase in the frequency of backache (4.6% vs. 6.8%, *p* < 0.047 *) and a significant decrease in irritability (62% vs. 57%, *p* < 0.032) and sleeping difficulties (34.1% vs. 26.9%, *p* < 0.001).

### 3.4. Symptoms of Anxiety and Depression

Symptoms of anxiety in adolescents were self-reported by 27.1% in 2021 and 27.2% in 2022, whereas depressive symptoms were reported by 15.4% in 2021 and 13.9% in 2022. For both indicators, the difference between the years was not statistically significant (*p* > 0.05).

Figure 2 shows the percentage of symptomatic patients according to their sex. In both years, girls were found to be significantly more often symptomatic than boys with symptoms of anxiety (2021, 34.6% vs. 19.2%; 2022, 34.7% vs. 19.2%; *p* < 0.001, respectively) and symptoms of depression (2021, 20.2% vs. 10.3%; 2022, 17.7% vs. 9.9%; *p* < 0.001, respectively). For both sexes, the difference between 2021 and 2022 was not statistically significant.

### 3.5. Predictors for HRQoL, Anxiety, Depressive Symptoms, and Psychosomatic Complaints

In a linear regression model, the sum scores of self- and proxy-reported HRQoL, self-reported anxiety (SCARED), self-reported depression (PHQ-2), and self- and proxy-reported psychosomatic complaints (HBSC) were explained separately for each timepoint, using possible predictors. Demographic variables from Table 1 and pandemic-related variables previously confirmed as predictors [15,28,29,30] were regarded as potential predictors. After controlling for the effects of predictors with and between sum scores, a stepwise regression model per sum score and timepoint was applied.

#### 3.5.1. Correlations with Demographic Variables

Spearman’s correlation coefficients for metric variables, age, and age × female sex were significantly correlated with all six sum scores (*p* < 0.001). For dichotomous variables, we tested for significant correlations with sum scores.

HRQoL: Self-reported HRQoL was significantly correlated with gender in 2021 (Cramer-V = 0.184, *p* = 0.003) and 2022 (0.165; *p* = 0.016), with migration background in 2021 (0.180; *p* = 0.02) and 2022 (0.173; *p* = 0.014), with single parenthood in 2021 (0.172; *p* = 0.026), and with living without balconies, terraces, and gardens in 2021 (0.232; *p* < 0.001). The proxy-reported HRQoL was significantly correlated with single parenthood in 2021 (0.111; *p* = 0.032) and 2022 (0.106; *p* = 0.017), with low parental education in 2022 (0.103; *p* < 0.001), and with living without balconies, terraces, and gardens in 2021 (0.151; *p* < 0.001).Anxiety: Self-reported symptoms of anxiety correlated significantly with sex in 2021 (0.251; *p* < 0.001) and 2022 (0.231; *p* < 0.001).Depressive symptoms: Self-reported depressive symptoms were significantly different for genders in 2021 (0.160; *p* < 0.001) and 2022 (0.127; *p* < 0.001), single parenthood in 2022 (0.088; *p* = 0.014), and living without balconies, terraces, and gardens in 2022 (0.087; *p* = 0.015).Psychosomatic complaints: Self-reported HBSC was significantly correlated with gender in 2021 (0.228; *p* < 0.001) and 2022 (0.206; *p* < 0.001), with low parental education in 2022 (0.164; *p* = 0.008), with living without balconies, terraces, or gardens in 2021 (0.182; *p* = 0.001), and with higher parental workload due to the pandemic in 2021 (0.200; *p* < 0.001) and 2022 (0.164; *p* = 0.015). Proxy-reported HBSC was found to differ significantly for gender in 2021 (0.113; *p* = 0.006) and 2022 (0.113; *p* = 0.001), migration background in 2021 (0.119; *p* = 0.004) and 2022 (0.111; *p* = 0.002), single parenthood in 2021 (0.107; *p* < 0.022) and 2022 (0.123; *p* < 0.001), low parental education in 2021 (0.114; *p* < 0.009), and living without balcony, terrace, or garden in 2021 (0.165; *p* < 0.001).

#### 3.5.2. Correlations with Pandemic-Related Variables

Pandemic-related predictors were obtained from adolescents’ self-reported HRQoL, symptoms of anxiety, depressive symptoms, and HBSC, and from parents’ reports of HRQoL and HBSC.

Parent-reported HRQoL was significantly correlated with pandemic-related predictors in 2021 and 2022: pandemic-related changes at school in 2021 (0.360; *p* < 0.001) and 2022 (0.350; *p* < 0.001); less contact with friends in 2021 (0.263; *p* < 0.001) and 2022 (0.284; *p* < 0.001); low family climate in 2021 (0.456; *p* < 0.001) and 2022 (0.421; *p* < 0.001); extended use of digital media in 2021 (0.284; *p* < 0.001) and 2022 (0.268; *p* < 0.001); children’s general burden due to the pandemic in 2021 (0.446; *p* < 0.001) and 2022 (0.407; *p* < 0.001); higher parental workload due to the pandemic in 2021 (0.191; *p* < 0.001) and 2022 (0.177; *p* < 0.001).Parent-reported psychosomatic complaints (HBSC) correlated significantly with pandemic-related predictors in 2021 and 2022: pandemic-related changes at school in 2021 (0.279; *p* < 0.001) and 2022 (0.299; *p* < 0.001); less contact with friends in 2021 (0.164; *p* < 0.001) and 2022 (0.187; *p*<0.001); low family climate in 2021 (0.397; *p* < 0.001) and 2022 (0.393; *p* < 0.001); extended use of digital media in 2021 (0.222; *p* < 0.001) and 2022 (0.219; *p* < 0.001); general children’s burden due to the pandemic in 2021 (0.387; *p* < 0.001) and 2022 (0.386; *p* < 0.001); higher parental workload due to the pandemic in 2021 (0.195; *p* < 0.001) and 2022 (0.176; *p* < 0.001).Self reported HRQoL correlated significantly with pandemic related predictors in 2021 and 2022: pandemic-related changes at school in 2021 (0.322; *p* < 0.001) and 2022 (0.338; *p* < 0.001); less contact with friends in 2021 (0.343; *p* < 0.001) and 2022 (0.316; *p* < 0.001); low family climate in 2021 (0.457; *p* < 0.001) and 2022 (0.409; *p* < 0.001); extended use of digital media in 2021 (0.235; *p* < 0.001) and 2022 (0.203; *p* < 0.001); children’s burden due to the pandemic in 2021 (0.418; *p* < 0.001) and 2022 (0.406; *p* < 0.001); higher parental workload due to the pandemic in 2021 (0.233; *p* < 0.001) and 2022 (0.174; *p* = 0.021).Self-reported symptoms of anxiety correlated significantly with pandemic related predictors in 2021 and 2022: pandemic-related changes at school in 2021 (0.233; *p* < 0.001) and 2022 (0.243; *p* < 0.001); less contact with friends in 2021 (0.204; *p* < 0.001) and 2022 (0.215; *p* < 0.001); low family climate in 2021 (0.274; *p* < 0.001) and 2022 (0.293; *p* < 0.001); extended use of digital media in 2021 (0.159; *p* < 0.001) and 2022 (0.161; *p* < 0.001); children’s burden due to the pandemic in 2021 (0.341; *p* < 0.001) and 2022 (0.346; *p* < 0.001); higher parental workload due to the pandemic in 2021 (0.155; *p* = 0.002) and 2022 (0.174; *p* < 0.001).Depressive symptoms correlated significantly with pandemic related predictors in 2021 and 2022: pandemic-related changes in 2021 (0.209; *p* < 0.001) and 2022 (0.238; *p* < 0.001); less contact with friends in 2021 (0.222; *p* < 0.001) and 2022 (0.245; *p* < 0.001); low family climate in 2021 (0.359; *p* < 0.001) and 2022 (0.334; *p* < 0.001); extended use of digital media in 2021 (0.143; *p* < 0.001) and 2022 (0.145; *p* < 0.001); general children’s burden due to the pandemic in 2021 (0.325; *p* < 0.001) and 2022 (0.314; *p* < 0.001); higher parental workload due to the pandemic in 2021 (0.12; *p* < 0.001) and 2022 (0.105; *p* = 0.002).Psychosomatic complaints correlated significantly with pandemic related predictors in 2021 and 2022: pandemic-related changes at school in 2021 (0.220; *p* < 0.001) and 2022 (0.254; *p* < 0.001); less contact with friends by 2021 (0.166; *p* = 0.013) and 2022 (0.216; *p* < 0.001); low family climate in 2021 (0.364; *p* < 0.001) and 2022 (0.366; *p* < 0.001); extended use of digital media in 2021 (0.219; *p* < 0.001) and 2022 (0.183; *p* < 0.001); children’s general burden due to the pandemic in 2021 (0.355; *p* < 0.001) and 2022 (0.345; *p* < 0.001); higher parental workload due to the pandemic in 2021 (0.200; *p* < 0.001) and 2022 (0.164; *p* = 0.015).

While demographic variables were not or only slightly correlated with each other and with pandemic-related variables, pandemic-related variables were correlated with each other (Kramer’s V between 0.1 and 0.3), as well as highly correlated with the variable “children’s general burden due to the pandemic”. Thus, we did not include this last predictor in the regression model (Kramer’s V = 0.358 in 2021 and 0.347 in 2022).

#### 3.5.3. Linear Regression Models

The results of the stepwise linear regression models are shown in Table 4 (the statistical analysis results are provided in Appendix A).

In both proxy-reported and self-reported surveys, key predictors of better HRQoL, anxiety, depressive symptoms, and psychosomatic symptoms were identified for 2021 and 2022. These predictors included factors such as age, gender, parental burden due to the pandemic, family climate, school burden, use of digital media, and contact with friends. Notably, some differences were observed between the predictors in 2021 and 2022, highlighting the changing nature of the factors influencing mental health outcomes over time.

#### 3.5.4. Linear Regression Diagnostics

For proxy-reported HRQoL, the homoscedasticity, independence, and normality of residuals were fulfilled. Durbin–Watson statistics were >1.9, and the variance inflation factor (VIF) was <1.2. Cook’s distance revealed some outliers, but the model fit the same with and without them. For self-reported HRQoL, multicollinearity was detected for gender and age; thus, gender was excluded from the model for 2021 and 2022. To account for autocorrelation, age was excluded in both years. After this correction, the Durbin–Watson statistic for both years was approximately 1.5. The VIF for all remaining variables was <1.2; the homoscedasticity, independence, and normality of residuals were fulfilled. Excluding outliers did not change the fit of the model.

For self-reported symptoms of anxiety, the homoscedasticity, independence, and normality of residuals were assessed. The Durbin–Watson statistic was >1.75, and the VIF was <1.2 for both years. Cook’s distance did not reveal outliers.

For self-reported symptoms of depression, the homoscedasticity, independence, and normality of residuals were fulfilled. The Durbin–Watson statistic was >1.75, and the variance inflation factor (VIF) was <1.2 for both years. Cook’s distance revealed some outliers; however, the model did not change after excluding them.

Sex was excluded because of self-reported psychosomatic symptoms owing to collinearity. With the remaining predictors, the homoscedasticity, independence, and normality of residuals were fulfilled. The Durbin–Watson statistic was >1.75, and the VIF was <1.2 for both years. Cook’s distance revealed some outliers; however, the model did not change after excluding them.

## 4. Discussion

This article discusses the levels and predictors of HRQoL, symptoms of anxiety, depression, and psychosomatic issues according to two surveys conducted in Italy in 2021 [28] and 2022, the results of which are reported here. This study aimed to compare youth problems during the second and third years of the pandemic in South Tyrol, Italy’s northernmost province. The withdrawal of restrictive measures in the context of pandemic control for the school and private lives of children and adolescents led to significant relief in 2022, when the survey was repeated, compared to 2021, which could also have been suitable for improving HRQoL and psychosocial conspicuousness. However, self-reported psychosomatic complaints did not decrease from 2021 to 2022, and the prevalence of different complaints varied between sexes and age groups. 

The COVID-19 pandemic has exacerbated existing mental health challenges and introduced new ones for children and adolescents, with factors such as social isolation, remote learning, and changes in family dynamics contributing to increased levels of anxiety and depression [41]. Gender differences in mental health outcomes during the pandemic have also been reported, with studies finding that girls are more susceptible to experiencing increased anxiety, depressive symptoms, and stress than boys [42]. The COVID-19 pandemic has had a major impact on the daily lives of children and young people. Isolation, home schooling, and social distancing are inevitable consequences of a global pandemic around the world. Several cross-sectional and longitudinal studies have examined the impact of the pandemic on the mental health of children and adolescents. Several narrative and systematic reviews, with and without meta-analyses, have been published, representing the current knowledge on the short and long-term effects of the pandemic on the mental health of the youngest generation [7,9,11,43,44,45,46,47,48,49,50,51,52,53,54,55]. The results were found to be highly variable and inconsistent around the world [7,51,52,54] with a general increase in mental health problems, especially anxiety and depression [11,44,45,47,48,49,52,55,56]. A much greater increase in mental health problems has been found in European children and adolescents than in children from Asia or other countries [11,43]. While Bussières et al. [43] showed that mental health problems in children up to 13 years of age increased three times more in European countries than in Asian countries during the pandemic, Deng et al. [11] found a prevalence of 31% for anxiety and depressive symptoms and 42% for sleep disturbance in children and adolescents worldwide, with higher percentages in the Eastern Mediterranean and European regions.

Depression and depressive symptoms in European youth were compared between the pre-pandemic and pandemic periods, and an increase in depression symptoms was found [48] including severe increases in mental disorders in Germany, Italy, and Poland [54]. Children and adolescents have reported deteriorated anxiety and depression levels after the COVID-19 pandemic [56]. In Asian countries, mainly in China, low economic status and living in rural areas were predictive factors influencing the mental health of children and adolescents during the pandemic [9,44,45,50], In European countries, school closures and significant problems within families [48,52] were predictive of increased mental health problems during the pandemic.

This study used longitudinal data to investigate the mental health and HRQoL during the COVID-19 pandemic. This study compared data from June 2021 and March 2022, and the predictors of HRQoL, anxiety, depressive symptoms, and psychosomatic complaints in 2022 differed from those in 2021. No significant differences were found in the prevalence of anxiety or depressive symptoms between 2021 and 2022. However, girls were found to be significantly more symptomatic than boys, with symptoms of anxiety and depression occurring in both years. Comparisons of self-reported psychosomatic complaints revealed no decrease between 2021 and 2022. Instead, a significant increase in headaches was found for boys, and parent proxy reports for adolescents indicate a significant increase in headaches from 2021 to 2022 for both boys and girls.

The study did not specifically compare the longitudinal development of mental health problems across different age groups [57]. However, comparable to other studies [58], it was found that, for younger children (7–10 years), parents reported significantly more sleeping difficulties in girls for both survey years, and they reported that boys felt significantly more nervous than girls in 2021 and 2022. Significant differences in HRQoL and psychosomatic complaints between genders have been reported [28,29,30]. In the present study, girls and their parents reported significantly lower HRQoL in 2021 than in 2022, and psychosomatic complaints differed significantly between sexes.

Several studies have been conducted in Germany [29,59,60,61]. Repeated waves of the national COPSY-Study [62] showed that mental health problems and worsening HRQoL almost doubled during the pandemic. Not all symptoms of mental health problems, such as depression, worsen [63]. Geweniger et al. [59] reported that mental health problems in children are strongly associated with pandemic-related variables, i.e., an increase in family conflict, inadequate social support, and the mental health of caregivers. In German-speaking Austria, Germany, Liechtenstein, and Switzerland, a significant proportion of children and adolescents experienced age-related mental health problems during the COVID-19 pandemic. Monitoring and longitudinal data collection were identified as important for addressing this situation [64,65,66].

In Italy, depression in children was found to be related to parental stress [67]. In the province of South Tyrol, increased rates of symptoms of anxiety, depression, and psychosomatic problems were observed in 2021, influenced by social distancing, parental stress due to the pandemic, and a low family climate [28]. These rates are highly dependent on age and single parenthood. Vicari and Pontillo [68] found an increased risk for mood disorders.

In our study, we observed a significant increase in HRQoL in 2022 compared with 2021, which could have been associated with the easing of restrictions and a gradual return to normal life. However, it is crucial to emphasize that HRQoL improvement does not necessarily translate to improvements in mental health symptoms, such as anxiety and depression. Notably, we found that the prevalence of these symptoms remained stable between the 2 years, with girls being more affected than boys. This highlights the need for further investigation into the factors contributing to the persistence of mental health symptoms, even after the relaxation of pandemic restrictions.

Moreover, our results revealed a substantial discrepancy between self-reported and proxy-reported psychosomatic complaints among adolescents, indicating that children and parents might perceive and interpret these issues differently. This underscores the importance of considering multiple perspectives when assessing youth mental health, and tailoring interventions accordingly. Furthermore, the persistence of psychosomatic complaints despite the easing of restrictions suggests that these issues may be closely related to other stressors and concerns, such as financial problems within the family. Therefore, it is essential to adopt a holistic approach when addressing the mental health needs of children and adolescents, taking into account various factors that may influence their wellbeing.

In summary, our findings contribute to the understanding of the complex interplay among pandemic-related restrictions, HRQoL, and mental health symptoms among children and adolescents. By identifying the nuances in these relationships, we can better inform targeted interventions and policies that address the specific needs of different age and gender groups, as well as account for the broader social determinants of mental health. This approach will be crucial in fostering resilience and promoting wellbeing among youth as we navigate the ongoing challenges and aftermath of the COVID-19 pandemic.

Although the results of meta-analyses have been inconsistent, general suggestions have been made to ensure an adequate supply of mental health services. Long-term monitoring is of high public health relevance [48,49], and health professionals should routinely screen patients for unmet mental health needs [8]. Long-term strategies need to be developed to identify and manage mental health problems caused by the pandemic [50,51,56]. These inconsistent results suggest that further well-designed studies and monitoring are needed [7,51,52,54]. In South Tyrol, the survey will be repeated by 2023, after the end of the pandemic.

Several factors could explain the lack of improvement in mental health at the second timepoint. The psychological impact of the pandemic, such as grief, trauma, stress related to illness, job loss, or financial instability, may persist even after restrictions are lifted. The transition back to regular routines and social interactions can be challenging for some individuals, particularly if they have become accustomed to isolation or remote learning during lockdown. At the time of the second survey, the pandemic had continued to evolve, with new variants and potential future waves of infection causing anxiety and uncertainty. To better understand the reasons for the lack of improvement in mental health at the second timepoint, further analyses should be conducted, including the impact of additional crises such as climate change and war [69]. By examining these potential explanations and conducting additional analyses, researchers can gain a deeper understanding of the factors that contribute to the persistence of mental health challenges, even after lifting pandemic restrictions.

Suggestions can be made concerning the longitudinal development of mental health problems and the identification of vulnerable groups at the local level. Firstly, our findings indicate that longitudinal data not only show the development of mental health symptoms, but also provide valuable insights into the changes in predictors over time. By comparing data from two different timepoints, we can better understand the evolving factors that contribute to mental health issues among children and adolescents and how these factors change in response to the pandemic and its subsequent effects on daily life. Secondly, in terms of differences between proxy-reported age groups (7–10 years and 11–19 years) in the longitudinal development of mental health problems, our study found that older adolescents experienced more psychosomatic complaints and mental health issues than younger children. This difference highlights the importance of considering age-specific factors and developmental stages when examining mental health trends and developing targeted interventions. Thirdly, our study identified groups that were more vulnerable to mental health problems at the local level. In particular, girls were found to be significantly more affected by anxiety and depression than boys were. Additionally, children and adolescents from families with a low family climate, single parenthood, and increased parental workload due to the pandemic were also at a higher risk of experiencing mental health issues. These findings emphasize the need for tailored support and interventions targeting these vulnerable groups to mitigate the negative impact of the pandemic on their mental wellbeing.

### Limitations

This study had some limitations. In addition to those discussed previously [28], the results showed that there were differences in the baseline characteristics of the survey participants between 2021 and 2022, including the average age of children and parents, lower educational level of parents, increased parental workload due to the pandemic, and living without a balcony, terrace, or garden. These differences could have limited the comparability of the two cohorts and introduced selection bias. The slight differences between the first and second surveys in the ages of the children and parents could be explained by the participants themselves. In the second survey, many emails were received from parents of children in their first year of school, who said that they did not feel able to fill in the questionnaire because their children had only been in school for 1 month. In the first survey, families with children in their final year of school did not feel responsible for filling in the questionnaire because their children had already left the school.

## 5. Conclusions

This study aimed to investigate the development of mental health and HRQoL in Italian children and adolescents in South Tyrol during the COVID-19 pandemic by comparing data from June 2021 and March 2022. We found that girls and their parents reported significantly lower HRQoL in 2021 than in 2022, whereas boys showed no significant change. Psychosomatic complaints differed significantly between sexes, and symptoms of anxiety and depression were reported by a considerable proportion of adolescents in both years, with girls being more symptomatic. By examining the mental health trends in this under-researched region, our study contributes to the development of targeted interventions that address the unique needs of specific populations and inform local and national policies.

Our research provides valuable data for policymakers and mental health professionals, enabling them to allocate resources and support more effectively to address the pressing mental health issues facing children and adolescents in the region. The lockdowns and home schooling in 2021 may have contributed to the observed differences between the two surveys. Our findings underscore the need for measures to improve the mental and physical health of children and adolescents after the pandemic, particularly for vulnerable groups, such as girls and those with pre-existing mental health conditions. Additionally, age, gender, and parental perceptions play crucial roles in mental health outcomes; thus, interventions should consider these factors.

In summary, this study highlights the significant impact of the COVID-19 pandemic on youth mental health, as well as the importance of ongoing monitoring to inform effective interventions and policies. Our study adds to the growing body of research on the mental health consequences of global crises by providing evidence of the differential effects of the pandemic on children and adolescents.

## Figures and Tables

**Figure 1 children-10-00895-f001:**
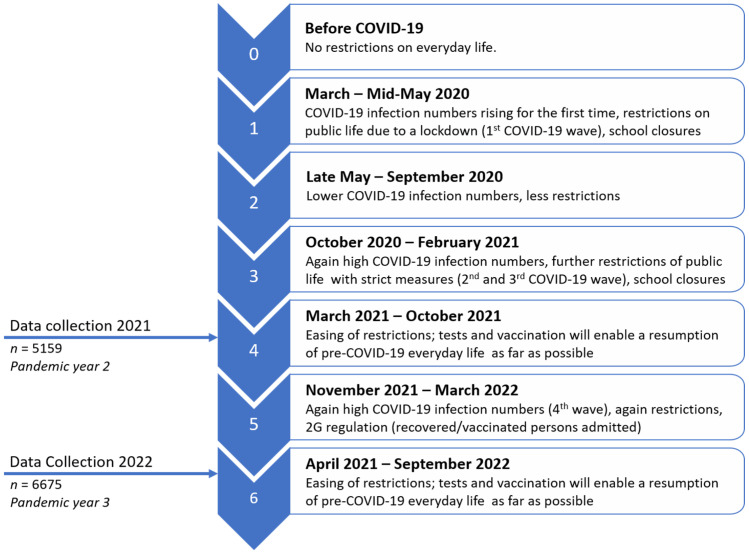
Pandemic stages and data collection timepoints. Reproduced with modification from Hansen et al. [16] under a Creative Commons Attribution noncommercial (unported, v3.0) license (http://creativecommons.org/licenses/by-nc/3.0/, accessed on 10 March 2023). Copyright © 2023, the authors. This reuse has not been endorsed by the licensor. The source reference is [16] and is available at https://www.mdpi.com/1660-4601/20/5/4478, accessed on 10 March 2023.

**Figure 2 children-10-00895-f002:**
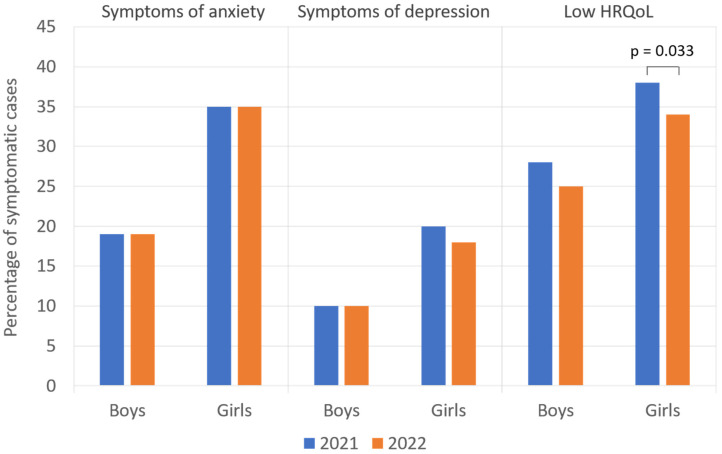
Adolescents’ reports of symptoms of anxiety, depression, and low health-related quality of life (HRQoL) per year and gender.

**Table 1 children-10-00895-t001:** Sociodemographic characteristics of children and adolescents aged 7–19 of the COP-S 2021 and COP-S 2022 samples.

	COP-S 2021		COP-S 2022		*p*-Value
	(*n* = 5169)		(*n* = 6665)		
	*n* (%)	M (SD)	*n* (%)	M (SD)	
Age		12.0 (3.58)		12.21 (3.58)	<0.001
7–10 years	2025 (39.2)		2432 (36.5)		0.003
11–19 years	3144 (60.8)		4233 (63.5)		
Gender					
Male	2560 (49.5)		3277 (49.2)		n.s.
Female	2609 (50.5)		3388 (50.8)		
Age of the parent, years		44.4 (6.4)		44.7 (6.2)	0.002
Gender of the parents, male	597 (11.5)		748 (11.2)		n.s.
Migration background					
No	4045 (88.4)		5200 (88.7)		n.s.
Yes	533 (11.6)		660 (11.3)		
Parental education					
Low	1134 (23.0)		1315 (20.6)		0.002
Moderate/high	3802 (77.0)		5078 (79.4)		
Single parenthood					
No	4603 (91.7)		6033 (91.2)		n.s.
Yes	419 (8.3)		579 (8.8)		
Live without balcony, terrasse, garden					
Yes	82 (1.6)		140 (2.2)		
No	4950 (98.4)		6318 (97.8)		0.038

M, mean; SD, standard deviation; n.s., not significant.

**Table 2 children-10-00895-t002:** HRQoL in children and adolescents aged 11–19 years in the second vs. third year of the COVID-19 pandemic, stratified by gender.

	Survey	Self-Reported			Parent-Reported		
		*n*	Low HRQoL ^1^	Normal/High HRQoL ^1^	*n*	Low HRQoL ^1^	Normal/High HRQoL ^1^
Boys	COP-S 2021	920	254 (27.6)	666 (72.4)	1280	391 (30.5)	889 (69.5)
	COP-S 2022	1004	251 (25)	753 (75)	1605	401 (25)	1204 (75)
	*p*-value ^2^		n.s.			0.001	
Girls	COP-S 2021	983	374 (38)	609 (62)	1324	443 (33.5)	881 (66.5)
	COP-S 2022	1076	361 (33.6)	715 (66.4)	1665	433 (26)	1232 (74)
	*p*-value ^2^		0.033			<0.001	
Boys and girls	COP-S 2021	1903	628 (33)	1275 (67)	2604	834 (32)	1770 (68)
	COP-S 2022	2080	612 (29.4)	1468 (70.6)	3270	834 (25.5)	2436 (74.5)
	*p*-value ^2^		0.015			<0.001	

^1^ Groups of low and normal/high HRQoL according to KIDSCREEN (for details, see Section 2). ^2^
*p*-values resulting from the χ^2^ test comparing the two groups of children and adolescents with low versus normal/high HRQoL across the COP-S 2021 and COP-S 2022 studies on self- and proxy-reported HRQoL. n.s., not significant.

**Table 3 children-10-00895-t003:** Subjective health complaints of children and adolescents experienced weekly or more often, according to scholars’ age, gender, and proxy or self-assessments.

	Age (Years)	Self (%)		Proxy (%)	
		Boys	Girls	Boys	Girls
Headache					
COP-S 2021	7–10	n.d.	n.d.	19.6	22.9
	11–19	32.0	47.8	28.2	39.4
	*p*-values ^1^	n.d./***		n.s./***	
COP-S 2022	7–10	n.d.	n.d.	22.6	28.0
	11–19	36.3	51.1	31.9	45.8
	*p*-values ^1^	n.d./***		**/***	
	*p*-values ^2^	n.d./*	n.d./n.s.	n.s./*	*/***
Abdominal pain					
COP-S 2021	7–10	n.d.	n.d.	25.6	29.6
	11–19	23.5	40.3	21.4	33.9
	*p*-values ^1^	n.d./***		n.s./***	
COP-S 2022	7–10	n.d.	n.d.	28.0	34.2
	11–19	26.7	41.7	22.7	36.3
	*p*-values ^1^	n.d./***		**/***	
	*p*-values ^2^	n.d./n.s.	n.d./n.s.	n.s./n.s.	*/n.s.
Backache					
COP-S 2021	7–10	n.d.	n.d.	4.6	6.3
	11–19	25.9	32.4	17.7	21.6
	*p*-values ^1^	n.d./**		n.s./*	
COP-S 2022	7–10	n.d.	n.d.	6.8	6.6
	11–19	25.5	33.1	18.3	23.4
	*p*-values ^1^	n.d./***		n.s./***	
	*p*-values ^2^	n.d./n.s.	n.d./n.s.	*/n.s.	n.s./n.s.
Feeling low					
COP-S 2021	7–10	n.d.	n.d.	28.8	26.3
	11–19	34.6	48.5	32.2	42.4
	*p*-values ^1^	n.d/***		n.s/***	
COP-S 2022	7–10	n.d.	n.d.	25.4	24.7
	11–19	35.0	50.8	35.1	44.9
	*p*-values ^1^	n.d./***		n.s./***	
	*p*-values ^2^	n.d./n.s.	n.d./n.s.	n.s./n.s.	n.s./n.s.
Dizziness					
COP-S 2021	7–10	n.d.	n.d.	5.4	5.5
	11–19	15.9	25.9	9.8	15.9
	*p*-values ^1^	n.d/***		n.s/***	
COP-S 2022	7–10	n.d.	n.d.	5.5	5.9
	11–19	16.8	25.0	11.0	15.9
	*p*-values ^1^	n.d./***		n.s./***	
	*p*-values ^2^	n.d./n.s.	n.d./n.s.	n.s./n.s.	n.s./n.s.
Irritable					
COP-S 2021	7–10	n.d.	n.d.	62.0	60.4
	11–19	55.7	66.8	64.5	69.5
	*p*-values ^1^	n.d./***		n.s./**	
COP-S 2022	7–10	n.d.	n.d.	57.0	56.1
	11–19	52.4	63.9	59.1	65.9
	*p*-values ^1^	n.d./***		n.s./***	
	*p*-values ^2^	n.d./n.s.	n.d./n.s.	*/**	n.s./*
Feeling nervous					
COP-S 2021	7–10	n.d.	n.d.	37.1	30.8
	11–19	39.6	49.6	42.0	42.8
	*p*-values ^1^	n.d/***		**/n.s	
COP-S 2022	7–10	n.d.	n.d.	35.0	28.5
	11–19	43.4	47.9	40.8	42.6
	*p*-values ^1^	n.d./*		**/n.s.	
	*p*-values ^2^	n.d/n.s.	n.d./n.s.	n.s./n.s.	n.s./n.s.
Sleeping difficulties					
COP-S 2021	7–10	n.d.	n.d.	34.1	38.8
	11–19	38.8	45.0	31.2	37.7
	*p*-values ^1^	n.d./**		n.s./*	
COP-S 2022	7–10	n.d.	n.d.	26.9	32.7
	11–19	36.8	43.7	31.7	35.2
	*p*-values ^1^	n.d./**		**/*	
	*p*-values ^2^	n.d./n.s.	n.d./n.s.	**/n.s.	**/n.s.

^1^ Boys vs. girls by age group (7–10 years/11–19 years). ^2^ June 2021 vs. March 2022 by reporting group (boys/girls and self/proxy). * *p* < 0.05, ** *p* < 0.01, *** *p* < 0.001. COP-S, “Corona and Psyche South Tyrol”; n.d., no data; n.s., not significant.

**Table 4 children-10-00895-t004:** Predictors of HRQoL and mental health in children and adolescents in the second and in the third year of the pandemic.

	HRQoL ^1^				Anxiety ^3,4^		Depressive Symptoms ^3,4^		Psychosomatic Complaints			
	Proxy-Report ^2^		Self-Report ^4^		COP-S 2021	COP-S 2022	COP-S 2021	COP-S 2022	Proxy-Report ^2,5^		Self-Report ^4,5^	
	COP-S 2021	COP-S 2022	COP-S 2021	COP-S 2022					COP-S 2021	COP-S 2022	COP-S 2021	COP-S 2022
Intercept	62.080 ***	60.944 ***	58.910 ***	54.914 ***	2.5192 ***	3.255 ***	−0.917 ***	−0.757 ***	41.000 ***	38.486 ***	41.643 ***	39.484 ***
Age	−0.334 ***	−0.370 ***					0.092 ***	0.090 ***	−0.178 ***		−0.264 ***	−0.127 *
Female												
Female × age			−0.175 ***	−0.120 ***	0.148 ***	0.117 ***	0.027 ***	0.018 ***	−0.068 ***	−0.061 ***	−0.145 ***	−0.117 **
Migration background			−1.716 *							−0.780 *		
Single parenthood	−1.826 *	−1.142 *							−1.114 *	−1.372 **		
Low parental education												0.661 *
Living without balcony, terrace, garden												
Parents’ burden due to pandemic	−1.597 ***	−1.068 **	−1.680 **		0.728 **	0.630 **	0.150 *		−1.084 ***	−0.613 **	−0.801 **	
Children’s burden due to situation at school	−4.439 ***	−4.281 ***	−3.730 ***	−4.472 ***	1.056 ***	1.210 ***	0.267 ***	0.360 ***	−1.030 ***	−1.372 **	−0.932 **	−1.527 ***
Less contact with friends	−3.234 ***	−3.060 ***	−4.376 ***	−3.833 ***	0.988 **	0.962 **	0.273 **	0.383 ***		−0.893 *		−0.510 *
Lower family climate	−7.680 ***	−6.899 ***	−7.369 ***	−6.874 ***	2.240 ***	2.611 ***	0.992 ***	0.929 ***	−3.076 ***	−2.508 ***	−3.556 ***	−3.722 ***
Extended use of digital media	−2.919 ***	−2.672 ***	−1.716 **									
Model fit (Adjusted R^2^)	0.285	0.255	0.275	0.224	0.161	0.162	0.202	0.178	0.183	0.156	0.163	0.167

The table shows the regression coefficients for all independent variables, controlling for the other predictors. * *p* < 0.05, ** *p* < 0.01, *** *p* < 0.005. ^1^ A higher value indicates better HRQoL. ^2^ Proxy report for children and adolescents 7–19 years of age. ^3^ A higher value indicates stronger symptoms and complaints. ^4^ Self-report of adolescents 11–19 years of age. ^5^ Higher values indicate fewer psychosomatic complaints.

## Data Availability

The data presented in this study are available from the corresponding author upon reasonable request.

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
