# Peer review of "Evolution of Youth’s Mental Health and Quality of Life during the COVID-19 Pandemic in South Tyrol, Italy: Comparison of Two Representative Surveys"

_children, 2023, doi:10.3390/children10050895_

Round 1

Reviewer 1 Report

Dear Authors, 

This paper is relevant to the field however I have some concerns. 

- Line 26- Can authors include the sample size in the abstract 

- Line 43-Can authors re-write the sentence. some phrases are repeated that affects the meaning. 

- Line 120- Although authors give a reference on the study design, I think authors should write out or summarize the study desgin, sampling etc. 

- Line 133- Can authors briefly describe the methodology 

-Line 136: Because authors did not  fully describe methods, its difficult to follow  most of the statements made in this sentence. Can authors address it 

- Line 167: The authors indicate that psychomotor problems in children were assessed by parents. Were the parents trained on how to administer this test, and how accurate were their test delivery? Can authors elaborate on this? 

- From this study , authors state that data was collected at two time points but authors fail to make a distinction in the write up on the exact timelines. Can authors be consistent with timeline statements in the text? . For instance in Line 191, I am not sure if baseline was between June 2021 and march 2022, because from Figure 1, he timelines for baseline survey is different

- In this study there is a baseline and endline assessment on 2 different populations and that seem flawed in the first place because, there may be differences in characteristics, that will affect mental health in the 2 groups at the 2 different time points. Did the authors do anything to mitigate this limitation? 

Line 366 to line 400: Looks like these belong to the methodology not results

Well written manuscript. Good presentation

Reviewer 2 Report

The paper reports on a representative, cross-sectional online survey investigating mental health outcomes among children and adolescents in an Italian region.

The topic of health literacy is highly relevant for various disciplines, including public health. However, the paper needs improvement on several issues:

1.

The text needs to be proofread by a native speaker, as there are several instances of grammatical errors in the current version.

2.

My main issue with the text is answering the “So what?” question. What are the key findings of the study to advance the literature on adolescents' mental health during the COVID-19 pandemic? Merely collecting data from an under-researched country, as stated by the authors, is not enough in my opinion.

3.

“Although there  was no decrease in self-reported psychosomatic complaints  584 from 2021 to 2022, there was a significant increase  in headaches among boys.”

The Discussion and Conclusion section needs the answer to the “Why” question. Why were there no improvements in mental health despite lifting restrictions etc., during the second time point? Is there any data/analysis that would provide this answer?

3.

The second main issue is that much space in the Results and Discussion section is given to several determinants of mental health, while a literature review of these determinants is not provided in the Theoretical part of the paper. Gender differences are just one such example.

4.

“This study has some limitations. In addition to those discussed previously  [28],  the  565 results show that there were differences in the baseline characteristics of the survey par- 566 ticipants between 2021 and 2022, including the average age of children and parents, lower  567 educational level of parents, increased parental workload due  to the pandemic, and living  568 without a balcony, terrace, or garden. These differences could limit the comparability of  569 the two cohorts and introduce selection bias.”

Is there a particular reason why authors did not control for these baseline differences, which would tease out the effect of time change in the analyses?

5.

“Our longitudinal study provides evidence of the differential effects of the COVID-19  588 pandemic on the mental health of children and adolescents. We found that lockdowns  589 and home schooling in 2021  may have contributed to the observed differences between  590  the two surveys.”

Does the data indicate that “lockdowns and homeschooling” contributed to the observed differences? Second, mental health did not largely change, so how did lockdown and homeschooling effect no change?

6.

The Discussion section largely repeats the Results. This section needs more contextualization and interpretation to answer the key “So what” question.

7.

What are the answers to questions b, c and d (page 3)?

8.

Considering the issues mentioned, I would advise revising and resubmitting.

Please, see above.

Reviewer 3 Report

Dear Authors,

Congratulations on your extensive work concerning  Evolution of Youth’s Mental Health and Quality of Life During the COVID-19 Pandemic in South Tyrol, Italy: Comparison of Two Representative Surveys

I suggest some minor revision:

Introduction

Since  research questions were formulated, some study hypotheses should be stated as well.

M&M:

How about adding a participant flowchart?

Data on the ethical issues (e.g. parental consent concerning children participating in the study) is missing.

Discussion

In the last part of discussion the authors should divide subsections: strenghts and limitations of the current study, future research implications, practical clinical implications..

The authors should also refer to study hypotheses.

Round 2

Reviewer 1 Report

Dear Authors, 

Thanks for taking the time to work on the comments. 

Well done

Reviewer 2 Report

The paper is improved.

The paper is improved with respect to the English language.

Author Response

Thank you for acknowledging the improvements made to the paper and the language. We apologize for any remaining minor language issues. We assure you that these issues will be thoroughly addressed during the proofreading process to ensure the highest quality and readability of the manuscript. We appreciate your valuable feedback and support throughout the revision process.